# Comorbid Cognitive Impairment in Chronic Obstructive Pulmonary Disease (COPD): Current Understanding, Risk Factors, Implications for Clinical Practice, and Suggested Interventions

**DOI:** 10.3390/medicina59040732

**Published:** 2023-04-08

**Authors:** Rayan A. Siraj

**Affiliations:** Department of Respiratory Care, College of Applied Medical Sciences, King Faisal University, Al-Ahsa 31982, Saudi Arabia; rsiraj@kfu.edu.sa

**Keywords:** COPD, cognitive impairment, cognition, dementia

## Abstract

Cognitive impairment is a common comorbidity in patients with COPD, significantly impacting health and clinical outcomes. Yet it remains under investigated and is largely overlooked. Although the exact cause of cognitive impairment in patients with COPD is still unclear, factors such as hypoxemia, vascular disease, smoking, exacerbation, and physical inactivity have been suggested. While international guidelines recommend identifying comorbidity in patients with COPD, such as cognitive impairment, cognitive assessment is not yet part of the routine assessment. Unidentified cognitive deficits in patients with COPD may have severe impacts on clinical management, resulting in an inability to maintain functional independence, poor self-management, and a greater dropout from pulmonary rehabilitation programs. There is a need to consider cognitive screening as a part of COPD assessment to promote early detection of cognitive impairment. Recognizing cognitive impairment early in the course of the illness allows the development of individualized interventions to meet patients’ needs and improve clinical outcomes. Pulmonary rehabilitation should be tailored to cognitively impaired patients with COPD to maximize the benefits and minimize the incompletion rate.

## 1. Introduction

Chronic obstructive pulmonary disease (COPD) is a common, non-communicable disease characterized by persistent respiratory symptoms and airflow limitation due to airway and/or alveolar abnormalities caused by smoking and air pollution [1]. COPD is preventable and treatable, although it remains not curable, which makes it a serious public health issue worldwide.

COPD is indeed a life-threatening disease with an increased mortality rate, numbering the third leading cause of death worldwide and accounting for 5% of all deaths worldwide. Statistics show that there were 455 million cases of COPD and 3.9 million deaths globally in 2019 [2]. Over the next few decades, COPD will be the cause of more than 5.4 million deaths annually.

Despite being a disease of the lungs, COPD comes with a wide range of extra-pulmonary manifestations, also known as comorbidities, including cardiovascular disease, diabetes, psychological disorders, and osteoporosis [3]. These comorbidities are not necessarily attributed to COPD, as they may occur on their own or as a result of shared risk factors. The typical profile of patients with COPD is that they coexist with comorbidities; evidence suggests that 98% of patients with COPD have either single or multiple comorbidities [4]. It is also worth mentioning that comorbidities increase the risk of mortality, and current evidence suggests that patients with COPD die more frequently from non-pulmonary causes compared with pulmonary causes [5,6].

Identifying and understanding the common comorbidities in COPD patients is important to provide better clinical judgment. Of the most common yet under-investigated comorbid conditions in COPD is cognitive impairment. This review will discuss various aspects related to cognitive impairment in COPD.

## 2. Cognition and Cognitive Impairment

Cognition refers to a set of brain functions in which one can understand and utilize new information through experience and thinking to adapt human behavior to a unique situation in a given environment [7]. Cognition encompasses several individual domains, including learning and memory, language, motor function, executive function, attention, and emotion. Within each cognitive domain, specific functional capabilities determine personal and intellectual skills (Table 1).

Cognitive impairment is a decline/dysfunction in one or more cognitive domains, such as memory, attention, and executive functioning [10]. Although mild cognitive impairment (MCI), which is the early stage of cognitive impairment, is noticeable, it is nevertheless less likely to impair functional dependency. Current evidence suggests that mild cognitive impairment may be considered the transitional state between the cognitive changes of normal aging and dementia [11], an irreversible disease with multiple impaired cognitive domains, despite not all patients with MCI progressing to dementia [10]. It is important to mention that in MCI, memory is more likely to be affected, and some people may experience subjective memory decline (how people feel about their memory), as opposed to other cognitive domains.

Indeed, evidence suggests that MCI is considered reversible and that a proportion of patients with MCI return to normal functioning [10]. Thus, it is important to understand any modifiable risk factors linked to MCI in order to prevent the progression to irreversible dementia.

The primary cause of the development of cognitive impairment is currently unknown. However, several factors have been suggested, including aging, smoking, diabetes, vascular disease, physical inactivity, infection, education, and genetics [12].

## 3. The Association between Lung Function and Cognitive Impairment

The current literature suggests that patients with lung impairment (e.g., chronic lung diseases) are at amplified risk of cognitive decline [13]. This could occur directly because of the airflow limitation or indirectly because of the shared risk factors in patients with lung diseases. Indeed, a number of studies examined the association between lung function measures and cognitive impairment in older people. Most of these studies support the link between lung function impairment and cognitive decline. For instance, in a large longitudinal study, which included more than 10,000 participants, the authors showed that subjects with impaired lung function at baseline were at greater risk of cognitive impairment [14]. Despite age being a main risk factor for cognitive impairment, there is evidence to suggest that lung function impairment is linked to cognitive impairment, irrespective of age [15]. It seems likely that lung impairment is an independent risk factor for cognitive impairment and rate of cognitive decline.

## 4. Cognitive Impairment in Patients with COPD

Although COPD and cognitive impairment have been studied independently, there is evidence to suggest the two conditions are interrelated. Since each condition is associated with poor clinical outcomes, such as increased mortality, it is likely that the risk increases in the presence of both conditions. There are common signs of cognitive impairment, which may impact patients’ independent functioning, self-management, and quality of life (Table 2).

Current prevalence estimates of cognitive impairment in COPD patients range from 4% to 61% [16,17]. The wider variations in these estimates are attributed to the study populations, assessment and diagnostic tools, and methodological designs [18,19]. A recent meta-analysis of 14 studies reported that, on average, 32% of patients with COPD are affected by cognitive impairment and that 1 in 4 people with COPD suffer from mild cognitive impairment [18]. This estimate is greater than that reported in the general population at 13%, highlighting a link between impaired lung function in patients with COPD and brain pathology, which cannot be disputed.

Studies have used various assessment methods to measure cognitive function in patients with COPD, leading to a wider variation in the estimated prevalence. For instance, using Mini-Mental State Examination (MMSE), a previous cross-sectional study of 940 patients with stable COPD reported the prevalence of cognitive impairment to be 39% [20]. Morris et al. reported that the prevalence of CI, using Addenbrooke’s Cognitive Examination (ACE)-III tools, was 50% (95% CI 33.8–66.3) in patients with COPD [21]. Even studies that used similar cognitive assessment tools had varying threshold definitions for cognitive impairment. The most commonly used cognitive test is MMSE, which is useful as a screening tool but not as a diagnostic tool [22].

COPD severity has been linked to a higher prevalence of cognitive impairment, although studies have defined COPD severity differently. For instance, in the nocturnal oxygen therapy (NOTT) trial, cognitive impairment was linked to the severity of COPD (measured with the presence of hypoxemia), ranging from 27% in mild hypoxemia to 62% in severe hypoxemia [17]. In addition, another study also showed there is an association between COPD severity and faster cognitive performance decline [23], indicating that those patients are prone to developing dementia. It should be noted, however, that the latter study defined COPD severity based on the use of supplemental oxygen rather than pulmonary function measurements. Using another severity measure, Siraj et al. also showed that patients with COPD and lower MRC scores (compared with higher MRC scores) were at an amplified risk of cognitive impairment [24]. Together, there appears to be a general consensus regarding how COPD severity affects cognitive impairment.

The link between increased risks of cognitive impairment in patients with COPD has also been demonstrated across several large-scale longitudinal studies. A recent meta-analysis involving six cohort studies also showed that COPD is linked with an amplified risk of cognitive impairment (Risk Ratio = 1.30, 95% CI = 1.13–1.49) [25]. Lutsey et al. demonstrated that midlife COPD was associated with a greater risk of cognitive impairment (odds ratio (OR): 1.33; 95% CI: 1.06–1.62) [26]. However, in patients who were diagnosed with COPD later in life, there was no association with cognitive impairment. This finding could be explained by survival bias (e.g., patients die before developing cognitive impairment) rather than the effect of COPD.

## 5. Hypothesized Mechanisms and Risk Factors

The exact etiology linking cognitive impairment to COPD remains unknown. The progressive airflow limitation, caused by airway and/or alveolar abnormalities, results in a ventilation–perfusion mismatch, eventually leading to chronic hypoxemia [27]. The lack of oxygen availability to the brain, which has a high oxygen demand, may put the brain at risk of developing ischemic damage. Indeed, impaired lung function results in inadequate oxygen supply to satisfy the needs of the brain, causing cerebral neuronal damage [28,29]. Therefore, hypoxia contributes to an increased risk of cognitive impairment by reducing the size and volume of the hippocampus [30].

Current evidence suggests various causal links between cognitive impairment and COPD, such as the neuronal damage mediated by hypoxia, and the affected oxygen-dependent enzymes essential for synthesizing neurotransmitters (e.g., acetylcholine), causing neuronal dysfunction [12]. Indeed, hypoxemia seems to be a significant risk factor in the development of cognitive impairment in patients with COPD, despite there being evidence to suggest that cognitive impairment is also prevalent in non-hypoxemic COPD patients. A meta-analysis, which included nine studies, demonstrated a negative association between partial pressure of oxygen (PaO2) and cognitive function in COPD patients [31]. Notably, deficits in cognitive function have also been observed in hypoxemic and non-hypoxemic patients with COPD, and cognitive performance is lower in patients with COPD (irrespective of the presence of hypoxemia) compared with healthy individuals [13]. This indicates that hypoxia may partly explain the causal relationship between cognitive impairment and COPD, but it is also important to consider other factors.

The extra-pulmonary manifestations associated with COPD, such as obstructive sleep apnea (OSA) and cardiovascular disease, also contribute to an increased risk of cognitive impairment among patients with COPD. Patients with COPD and OSA suffer from decreased oxygen saturation, which has serious impacts on sleep quality (e.g., sleep fragmentation), leading to specific cognitive deficits such as attention, memory, executive function, and language abilities. In addition, patients with both COPD and OSA are also prone to profound hypoxemia [32], increasing the risk of cognitive dysfunction.

Moreover, cardiovascular disease, which is among the most common comorbidities in patients with COPD, also contributes to a greater risk of cognitive impairment [33]. Indeed, each disease by itself (COPD and CV disease) is associated with a greater risk of cognitive impairment; hence, it is likely that in the co-existence of both diseases, the risk of cognitive impairment is magnified. It is worth mentioning that cognitive impairment is a consequential effect of small cerebral vessels, and neuroimaging studies showed that cerebrovascular damage plays a crucial role in the development of cognitive impairment in patients with COPD [34]. Furthermore, COPD and CV disease share similar risk factors, such as smoking and hypoxemia, which have been shown to be linked to cognitive impairment [35]. Therefore, the CV comorbidity in patients with COPD contributes to cognitive impairment in patients with COPD by reducing cerebral perfusion, leading to neural damage.

Systemic inflammation, and oxidative and physiological stress have also been proposed as plausible mechanisms contributing to cognitive impairment in COPD [36,37]. To explain, patients with COPD have increased levels of systemic inflammatory markers such as C-reactive protein, interleukin 6, and fibrinogen, all of which are associated with an increased risk of cognitive impairment [35]. Further, low-grade systematic inflammation has been associated with lower cognition levels [38,39]. It is also worth mentioning that patients with COPD, especially at an advanced stage, are at increased risk of cardiovascular disease (CVD) [40], thus contributing to an increased risk of cognitive impairment.

Another key factor contributing to cognitive impairment in patients with COPD is inflammation, defined as the development of low-grade inflammation because of aging [41]. It is well known that aging is a major risk factor for cognitive impairment and dementia, and there is evidence to show that the prevalence of cognitive impairment increases as people age [42]. In the situation where older people have COPD (both aging and COPD are associated with systemic inflammation), the risk of cognitive impairment in this population is likely to increase. Indeed, a recent large-scale population-based study shows that the incidence of cognitive impairment in COPD is greater than in matched subjects without COPD [24].

The role of smoking cannot be excluded, as it increases the risk of COPD and vascular brain damage, leading to cognitive impairment. Indeed, evidence suggests smoking may cause cognitive decline by exacerbating cerebral hypoxia through extended exposure to carbon monoxide, shifting the oxyhemoglobin dissociation curve to the left [43]. In a large multicenter cohort study, which included 17,610 participants, an annual decline in cognitive functioning was associated with smoking, even after adjusting for confounders [44]. Despite smoking being an independent risk factor for cognitive dysfunction, there is evidence to show that lung function impairment is associated with cognitive impairment, irrespective of smoking [45]. Thus, it is less likely that smoking alone would explain the relationship between COPD and cognitive impairment.

Exacerbations of COPD are significant events in the course of the disease and have been linked to worse clinical outcomes, such as hospital admission and mortality [46]. Exacerbations are associated with infections, hypoxemia, and physiological and psychological stress. In addition, COPD exacerbations have been linked to a faster lung function decline, even if at a single event [47]. Therefore, it is reasonable to assume that exacerbation may influence cognitive functioning in COPD patients. Baja et al. showed that COPD patients, during exacerbations, had worse cognitive impairment compared with patients with heart failure [48]. Although there is evidence to show that acute cognitive dysfunction improves after six weeks [49], it is nevertheless unknown whether exacerbation influences cognitive abilities in the long term [50].

Increasing evidence suggests that there is a positive association between increased physical activity and overall health in patients with COPD [35]. Indeed, people who regularly exercise are less prone to systemic inflammation and oxidative stress; both systemic inflammation and oxidative stress are considered risk factors for cognitive impairment in patients with COPD [51]. In addition, exercise has also been linked to alleviating psychological disorders, such as anxiety and depression, which are common in patients with COPD [52] and have been shown to increase the risk of cognitive impairment [53]. Although a number of studies have been conducted on the effect of exercise on cognition in patients with COPD, there has not been a meta-analysis carried out to show the robustness of the current evidence. Therefore, the effect of exercise, particularly pulmonary rehabilitation, on improving cognitive performance in COPD patients is to be ascertained.

## 6. Clinical Implication of Cognitive Impairment in COPD

The presence of cognitive impairment has serious impacts on patients with COPD in many aspects of clinical management, such as functional dependence, adherence to medications, smoking cession, and pulmonary rehabilitation completion rate.

Impaired executive function, learning, memory, and visual processing may potentially interfere with self-management. Indeed, self-management has been demonstrated to enhance dyspnea and health-related quality of life, as well as decrease causes of hospitalizations, and is, therefore, crucial in the management of COPD [54]. Self-management, which requires a complex cognitive process, involves symptom monitoring, adhering to medications, maintaining a healthy lifestyle (healthy diet and physical activity), and coping with daily living [55]. Therefore, patients with COPD must have adequate cognitive functioning in order to maintain functional independence, and follow medication instructions, all of which are essential components of effective care. However, in the presence of cognitive dysfunction, self-management may likely be compromised in COPD patients.

There has been little research on the impact of cognitive impairment on self-management in COPD patients. However, a previous systematic review of 13 studies found that cognitively impaired COPD patients require more daily living assistance, are less likely to adhere to treatment, and are unable to use inhalers correctly [56], all of which contribute to ineffective self-management. Indeed, proper inhaler technique, which is key for effective self-management and important to optimize drug delivery [57], is significantly affected in patients with cognitive impairment. Current evidence also shows that a score of 23–24 or less in the MMSE is predictive of poor inhaler technique [56]. Further, impaired executive function and praxis (planning and sequencing) were also linked to improper inhaler technique [58]. It is nevertheless important to note that some inhaler devices (metered dose inhalers) are more difficult to use than others (Turbohalers); thus, it is important to consider the presence of cognitive impairment when prescribing inhalers. To date, however, no recommendations on the use of inhalers in patients with COPD and cognitive impairment are currently available.

Pulmonary rehabilitation (PR) is considered the cornerstone for COPD management and aims to improve clinical outcomes as well as reduce hospitalization and mortality. The PR program involves exercise training, education, behavioral change, and nutritional support; all of which require adequate cognitive capabilities [59]. However, the presence of cognitive impairment poses negative impacts on compliance and adherence to PR. A previous cross-sectional study of 157 individuals with COPD found that COPD patients with cognitive impairment had a higher dropout rate (23%) compared with patients without cognitive impairment (10%) [60], suggesting that cognitive impairment could interfere with patients’ ability to complete PR. Indeed, completing a PR program requires several cognitive functions, such as analyzing, planning, and controlling impulses, all of which may be affected by cognitive impairment, leading to a greater dropout rate.

## 7. Assessment of Cognitive Impairment in COPD

Appropriate intervention strategies start with a timely recognition of cognitive dysfunction in patients with COPD. Indeed, assessing cognitive function is not a complex procedure and does not require advanced equipment. It can be carried out using one of the validated screening tools, which can give an overview of whether patients with COPD have signs of cognitive impairment and should therefore be referred to a specialist for a comprehensive assessment [61].

Underdiagnosed cognitive impairment, especially in the early stage of cognitive decline (mild cognitive impairment), has severe impacts on the clinical outcomes of patients with COPD. Screening those patients may potentially prevent patients’ deterioration and allow for early intervention. Several validated screening tools are available that can be suited for routine clinical practice. Some of these tools are freely available, while others require a licensed practitioner to be used.

Clinically, it is essential to have a screening tool that is short, easy to use, accurate, and assesses many cognitive domains. A variety of tools are available for assessing cognitive impairment; some tools are comprehensive (if they measure each specific domain; please refer to Table 1), while others are non-comprehensive (if they assess a limited spectrum of cognitive impairment). The duration of the assessment varies, ranging from 5 to 10 min, depending on the assessment tools used.

Mini-Mental State Examination (MMSE) remains the most common screening tool for cognitive impairment [61], although it may underestimate impairment compared with other cognitive tests. In fact, there is evidence to suggest that MMSE is not the best screening tool available to detect MCI. Another cognitive assessment test is the Montreal Cognitive Assessment (MoCA) [62], which has been shown to detect MCI in COPD patients. However, it should be noted that screening for cognitive function, particularly MCI, in COPD patients should be conducted using a comprehensive battery. There are also other cognitive tests available such as the Addenbrooke’s Cognitive Examination (ACE) [63], the Clock Drawing Test (CDT) [64], the Mini-Cog test [65], Informant Questionnaire on Cognitive Decline in the Elderly (IQCODE) [66], Cambridge Cognitive Examination (CAMCOG) [67], Quick Cognitive Screening Test (QCST) [68], Memory and Executive Screening (MES) [69], and Virtual Reality Day-Out-Task (VR-DOT) [70] (Table 3). It is also important to mention these are not the only available ones; other tests can be found in the literature [71].

With the existence of a wide variety of screening tools, healthcare professionals may need help selecting which tools to use and the appropriate method of administration. It is likely that utilizing a disease-specific cognitive impairment tool (if available) is the first choice as it provides a better picture of the cognitive status [22]. Alternatively, domain-specific screening measures would be the second option. It is likely that the accuracy of recognizing the presence of cognitive impairment may increase if information from the patients is combined with an informant questionnaire with domain-specific measures [71]. 

While cognitive impairment, on the broader spectrum, is common in patients with COPD, the role of screening during routine visits remains unclear. In other words, whether all patients with COPD should undergo a cognitive assessment or if it should be restricted to a group of patients presenting with cognitive dysfunction symptoms is to be ascertained.

## 8. Suggested Intervention for Managing Cognitive Impairment in Patients with COPD

Since cognitive impairment is associated with considerable negative impacts on patients with COPD, action should be taken within the frame of COPD management. Modifiable risk factors linked to increased prevalence and incidence of cognitive impairment, such as smoking, hypoxemia, and comorbidities, should therefore be addressed. Several interventions have been suggested that might improve cognitive function in patients with COPD.

Hypoxemia is a major risk factor for developing cognitive impairment in patients with COPD. While oxygen therapy is a well-established treatment for hypoxemia, the role of supplemental oxygen in improving cognitive functioning in patients with COPD remains uncertain. While studies on the impact of acute oxygen therapy showed no beneficial effect, long-term oxygen therapy may have minimal effect. For instance, Thakur et al. showed that continuous oxygen therapy was associated with a reduced risk of cognitive impairment in patients with COPD [28]. However, the latter study was cross-sectional in nature; therefore, it could not establish whether oxygen therapy would improve cognitive performance. Long-term oxygen therapy seems to improve functioning in COPD, but it might not prevent it completely. More studies are required to ascertain this observation.

While there is some evidence to suggest that cognitive training is beneficial in improving some cognitive domains in patients with MCI, data on patients with COPD are currently lacking. A previous randomized controlled trial (RCT) aimed to investigate whether cognitive training improved attention, learning, and logical–deductive thinking in a group of 105 hypoxemic patients with COPD [72]. Patients were allocated into an intervention group (multidimensional care with cognitive training) and a control group (multidimensional care without cognitive training). The findings showed that some cognitive domains improved in both groups, but no additional improvements were observed in the intervention group.

Pulmonary rehabilitation (PR) is a multidisciplinary intervention offering exercise training and education for patients with COPD. It is a cornerstone in managing COPD patients to alleviate symptoms and improve physical activity and overall health. Current evidence suggests that the exercise training incorporated within the PR program may be beneficial to improve cognitive disorders. However, no conclusive evidence exists that exercise improves cognitive impairment in COPD patients. In a prospective 3-month PR trial, which included 34 patients with COPD, Pereira et al. reported significant improvement in cognitive performance even after accounting for confounders [73]. In addition, Kozora et al. reported improvement in specific cognitive domains such as visual attention, verbal memory, and visuospatial skills in a group of COPD patients who completed a 3-week PR program [74]. Nevertheless, due to the variability of exercise training cognitive outcome measures, it remains difficult to ascertain whether exercise and PR will improve cognitive function in this population. This opens new doors for research to investigate this in the future.

Despite the significant benefits of PR programs among patients with COPD, there is a need to tailor PR programs for those with cognitive impairment. This is because PR requires functional dependence and adherence to instruction for optimal benefits. However, in case of severe cognitive impairment in patients with COPD, patients are less likely to adhere to instructions, making it challenging to obtain the benefits of PR. Therefore, there is a need to assess the appropriate model for PR for those patients. A new approach to evaluate the PR program in patients with COPD is through metabolomics and proteomics. A recent study conducted by Maniscalco and colleagues, which included 40 subjects (35 in the experimental group and 15 in the control group), showed that nuclear magnetic resonance (NMR)-based metabolomics of exhaled breath condensate could be used to monitor PR in patients with COPD, as it provides an unbiased tailored approach [75]. However, further research would be worthwhile.

Lung volume reduction surgery has been proposed as an intervention for neuropsychological measures. In an older trial, patients with emphysema who had undergone LVRS showed significant improvement in neuropsychological measures (e.g., anxiety and depression) and cognitive functions (e.g., sequential skills and verbal memory) [37]. However, it is important to mention that the latter study was relatively small, involving 20 subjects in the control group and 19 patients in the experimental group, meaning that despite the improvement in specific cognitive functions, this area requires in-depth investigation.

## 9. Conclusions

While cognitive impairment is prevalent among patients with COPD, it is largely overlooked, imposing a substantial burden on health and prognosis. There is an unmet need to better understand the consequences of cognitive impairment in COPD patients in order to provide targeted interventions. Healthcare professionals in direct contact with patients with COPD should be observant of cognitive deficits and know how to screen for those patients. Early identification of cognitive impairment allows appropriate intervention and prevents delayed diagnosis, all of which aim to improve patients’ outcomes.

## Figures and Tables

**Table 1 medicina-59-00732-t001:** Main domains and their cognitive functions [8,9].

Cognitive Domain	Cognitive Function
Learning and Memory	Recall Short-term memory Long-term memory Reasoning memory Information processing
Language	Fluency Naming objects Wording Language receptive
Motor function	Coordination Visual perception Visuoconstruction
Executive Function	Planning Organizing Decision making Reasoning Working memory Responding to feedback
Attention	Processing speed Sustained attention Alternating attention
Emotion	Self-evaluation Self-control Recognizing emotions

**Table 2 medicina-59-00732-t002:** Common signs of cognitive impairment.

Signs of Cognitive Impairment
Loss of memory. Need help finding a word. Asking the same questions repetitively. Inability to recognize family members, places, events, dates, or days. Difficulty making a decision—loss of executive functioning. Difficulty maintaining functional independence. Frequent mood swings. Difficulty seeing objects that is not caused by a vision problem.

**Table 3 medicina-59-00732-t003:** Assessment tools for cognitive impairment.

Cognitive Assessment Tools	Description
Mini-Mental State Exam (MMSE)	It is the most common cognitive assessment tool. It takes up to 10 min to complete. It involves counting backward and identifying objects.
Montreal Cognitive Assessment (MoCA)	This test involves performing several tasks, such as naming objects, drawing, memorizing words, and copying shapes. It takes up to 15 min to complete.
Addenbrooke’s Cognitive Examination (ACE)	A brief cognitive test that aims to assess different cognitive domains. It takes up to 5 min to administer.
Clock Drawing Test (CDT)	It is a nonverbal tool, asking the patients to draw a clock and a hand to indicate the time. It may be sensitive to dementia. It is better than MMSE for detecting cognitive impairment.
Mini-Cog test	It aims to assess memory and consists of a three-word recall test that is added to CDT. The inability to recognize the three words is a sign of cognitive impairment.
Informant Questionnaire on Cognitive Decline in the Elderly (IQCODE)	It is a simple cognitive measure for specific domains (memory, verbal intelligence, and performance). It takes between 15–20 min to administer.
Cambridge Cognitive Examination (CAMCOG)	It is a comprehensive assessment tool that takes up to 30 min to complete. It focuses on multiple cognitive domains such as orientation, attention, and short- and long-term memory.
Quick Cognitive Screening Test (QCST)	A short and brief cognitive test that assesses several cognitive domains. It takes between 8–15 min.
Memory and Executive Screening (MES)	A comprehensive neuropsychological assessment tool that assesses memory, language, attention, and executive function.
Virtual Reality Day-Out-Task (VR-DOT)	It is used as a screening tool for early dementia, which focuses on assessing functional impairment.

## Data Availability

Not applicable.

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
