# Peer review of "Comorbid Cognitive Impairment in Chronic Obstructive Pulmonary Disease (COPD): Current Understanding, Risk Factors, Implications for Clinical Practice, and Suggested Interventions"

_medicina, 2023, doi:10.3390/medicina59040732_

Round 1

Reviewer 1 Report

The paper by Rayan A. Siraj (“Comorbid Cognitive Impairment in Chronic Obstructive Pulmonary Disease (COPD): Current Understanding, Risk Factors, Implication for Clinical Practice and Suggests Interventions”) deals with cognitive impairment in COPD patients, an underestimated comorbidity.

The paper is interesting but presents several problems.

1. The references are quite old: 18 are published before 2010; 25 between 2010 and 2014; 16 between 2015 and 2019; and only 5 from 2020 on. They should be upgraded to more recent data.

2. Regarding the general description of dementia, reference to papers describing dementia in COPD should be avoided. Instead, the author should refer to studies describing dementia only, and report details on subjective memory decline (SMD) and mild cognitive impairment (MCI).

3. When discussing systemic inflammation (page 5, line 151), the author should introduce the inflammaging (for example, Nat Rev Endocrinol 2018;14, 576), a persistent and general inflammation state affecting ageing and age-related diseases.

4. Papers referring to the relationship cardiovascular disease (line 157) and cognitive impairment should be included.

5. New approaches using metabolomics and proteomics to evaluate pulmonary rehabilitation should also be considered (for example, Cells 2022;11(3):344) since biomarkers can help in the evaluation of dementia in COPD.

Author Response

Response to Comments from Reviewer 1

Comment 1

The references are quite old: 18 are published before 2010; 25 between 2010 and 2014; 16 between 2015 and 2019; and only 5 from 2020 on. They should be upgraded to more recent data

Response 1:

References are updated where needed.

Comment 2:

Regarding the general description of dementia, reference to papers describing dementia in COPD should be avoided. Instead, the author should refer to studies describing dementia only, and report details on subjective memory decline (SMD) and mild cognitive impairment (MCI).. 

Response 2:

Thank you. I have revised the referenced paragraph to the following (page 5; lines 73-82) “ Cognitive impairment is a decline/dysfunction in one or more cognitive domains, such as memory, attention, and executive functioning [1]. Although mild cognitive impairment (MCI), which is the early stage of cognitive impairment, is noticeable, it is nevertheless less likely to impair functional dependency. Current evidence suggests that mild cognitive impairment may be considered the transitional state between the cognitive changes of normal aging and dementia  [2], an irreversible disease with multiple impaired cognitive domains, despite not all patients with MCI progress to dementia [1]. It is important to mention that in MCI, memory is more likely to be affected and some people may experience subjective memory decline (how people feel about their memory), as opposed to other cognitive domains.

Comment 3:

When discussing systemic inflammation (page 5, line 151), the author should introduce the inflammation (for example, Nat Rev Endocrinol 2018;14, 576), a persistent and general inflammation state affecting aging and age-related diseases.

Response3:

Thank you for the opportunity to elaborate. I have added the following paragraph (page 12; lines 202-210) “Another key factor contributing to cognitive impairment in patients with COPD is inflammaging, defined as the development of low-grade inflammation because of aging [3]. It is well known that aging is a major risk factor for cognitive impairment and dementia, and there is evidence to show that the prevalence of cognitive impairment increases as people age [4]. In the situation where older people have COPD (both aging and COPD are associated with systemic inflammation), the risk of cognitive impairment in this population is likely to increase. Indeed, a recent large-scale population-based study shows that the incidence of cognitive impairment in COPD is greater than matched subjects without COPD [5].”   

Comment 4:

Papers referring to the relationship between cardiovascular disease (line 157) and cognitive impairment should be included.

Response 4:

This has been addressed.

Comment 5:

New approaches using metabolomics and proteomics to evaluate pulmonary rehabilitation should also be considered (for example, Cells 2022;11(3):344) since biomarkers can help in the evaluation of dementia in COPD.

Response 5:

Thank you. I have added the following paragraph (page 18; lines 362-372) “ Despite the significant benefits of PR programs among patients with COPD, there is a need to tailor PR programs for those with cognitive impairment. This is because PR requires functional dependence and adherence to instruction for optimal benefits. However, in case of severe cognitive impairment in patients with COPD, patients are less likely to adhere to instructions, making it challenging to get the benefits of PR. Therefore, there is a need to assess the appropriate model for PR for those patients. A new approach to evaluate the PR program in patients with COPD is through metabolomics and proteomics. A recent study conducted by Maniscalco and colleagues, which included 40 subjects (35 in the experimental group and 15 in the control group) showed that nuclear magnetic resonance (NMR) -based metabolomics of exhaled breath condensate could be used to monitor PR in patients with COPD, as it provides unbiased tailored approach [6]. However, more research is worthwhile.

Reviewer 2 Report

The manuscript titled “Comorbid Cognitive Impairment in Chronic Obstructive Pulmonary Disease (COPD): Current Understanding, Risk Factors, Implication for Clinical Practice and Suggests Interventions” discusses about Cognitive impairment in patients with COPD and elucidates various factors which can attribute to it.

Though, the review is well-written but is not exhaustive enough to provide complete landscape of research and various hypothesis associated in the field.

Some areas of improvement are:

1.      A key mechanism proposed for cognitive dysfunction in COPD is neuronal damage mediated through hypoxia, but it has also been suggested that oxygen-dependant enzymes which are important in the synthesis of neurotransmitters, such as acetylcholine, may be affected, which should be discussed in the manuscript.

2.      Author needs to elaborate on various causes and tools for cognitive impairment in the article.

3.      Author should also discuss on the effect of Lung Volume Reduction Surgery in cognitive impairment in suggested interventions (Section 8).

4.      There are numerous typographical errors like “Self-evaluation”, “Emotion” etc in Table 1

5.      Redundant sentences for example-lines 29 and 33, lines 220 and 289. Repetition can be avoided. Similarly, Section 3 (line 72) and Section 4 (line 85) can be merged as in both sections association of COPD and cognitive impairment have been described.

6.      In Reference section, the numbering is appearing twice in each reference.

Author Response

Response to Comments from Reviewer 2 

Comment 1:

A key mechanism proposed for cognitive dysfunction in COPD is neuronal damage mediated through hypoxia, but it has also been suggested that oxygen-dependant enzymes which are important in the synthesis of neurotransmitters, such as acetylcholine, may be affected, which should be discussed in the manuscript.

Response 1:

Thank you for the comment. I have added a key mechanism linking COPD to cognitive impairment, see (page 10; lines 161173) “ Current evidence suggests various causal links between cognitive impairment and COPD, such as the neuronal damage mediated by hypoxia, and the affected oxygen dependent-enzymes essential for synthesizing neurotransmitters (e.g. acetylcholine), causing neuronal dysfunction [7]. Indeed, hypoxemia seems to be a significant risk factor in the development of cognitive impairment in patients with COPD, despite there is evidence to suggests that cognitive impairment is also prevalent in non-hypoxemic COPD patients. A meta-analysis, which included nine studies, demonstrated a negative association between partial pressure of oxygen (PaO2) and cognitive function in COPD patients [8]. Notably, deficits in cognitive function have also been observed in hypoxemic and non-hypoxemic patients with COPD and cognitive performance is lower in patients with COPD (irrespective of the presence of hypoxemia) compared to healthy individuals  [9, 10]. This indicates that hypoxia may partly explain the causal relationship between cognitive Impairment and COPD, but it is also important to consider other factors.

Comment 2:

Author needs to elaborate on various causes and tools for cognitive impairment in the article.

Response 2:

Thank you for the opportunity to elaborate. I have added the following paragraphs in section 5 (pages 10-12; lines 161-192)”  Current evidence suggests various causal links between cognitive impairment and COPD, such as the neuronal l damage mediated by hypoxia, and the affected oxygen dependent-enzymes essential for synthesizing neurotransmitters (e.g. acetylcholine), causing neuronal dysfunction [7]. Indeed, hypoxemia seems to be a significant risk factor in the development of cognitive impairment in patients with COPD, despite there is evidence suggests that cognitive impairment is also prevalent in non-hypoxemic COPD patients. A meta-analysis, which included nine studies, demonstrated a negative association between partial pressure of oxygen (PaO2) and cognitive function in COPD patients [8]. Notably, deficits in cognitive function have also been observed in hypoxemic and non-hypoxemic patients with COPD and cognitive performance is lower in patients with COPD (irrespective of the presence of hypoxemia) compared to healthy individuals  [9, 10]. This indicates that hypoxia may partly explain the causal relationship between cognitive Impairment and COPD, but it is also important to consider other factors.

The extra-pulmonary manifestations associated with COPD, such as obstructive sleep apnea (OSA) and cardiovascular disease, also contribute to an increased risk of cognitive impairment among patients with COPD. Patients with COPD and OSA suffer from decreased oxygen saturation, which has serious impacts on sleep quality (e.g. sleep fragmentation), leading to specific cognitive deficits such as attention, memory, executive function and language abilities. In addition, patients with both COPD and OSA are also prone to profound hypoxemia [11], increasing the risk of cognitive dysfunction increasing the risk of cognitive dysfunction.

Moreover, cardiovascular disease, which is among the most common comorbidities in patients with COPD, also contributes to a greater risk of cognitive impairment. Indeed, each disease by itself (COPD and CV disease) is associated with a greater risk of cognitive impairment; hence, it is likely that in the co-existence of both diseases, the risk of cognitive impairment is magnified. It is worth mentioning that cognitive impairment is a consequential effect of small cerebral vessels; nuroimaganing studies showed that cerebrovascular damage plays a crucial role in the development of cognitive impairment in patients with COPD [12]. Furthermore,  COPD and CV disease share similar risk factors, such as smoking and hypoxemia, which have been shown to be linked to cognitive impairment. Therefore, the CV comorbidity in patients with COPD contributes to cognitive impairment in patients with COPD, by reducing cerebral perfusion, leading to neural damage.

In section 7, I added the following paragraph (page 16; lines 295-300)” Clinically, it is essential to have a screening tool that is short, easy to use, accurate, and assesses many cognitive domains. A variety of tools are available for assessing cognitive impairment; some are comprehensive (if they measure each specific domain; please refer to Table 1), while others are non-comprehensive (if they assess a limited spectrum of cognitive impairment). The duration of assessment varies, ranging from 5 to 10 min, depending on the assessment tools used”.

On page 17; lines 311-317, I added the following “  With the existence of a wide varaity of screening tools, healthcare professianls may find it difficult to select which tools to use and the appropriate method of administration. It seems likely that utilizing a disease-specific cognitive impairment tool (if available), should be the first choice as it provides a better picture of the cognitive status. Alternatively, domain-specific screening measures would be the second option. It is likely that the accuracy of recognizing the presence of cognitive impairment may increase if information from the patients is combined with an informant questionnaire if domain-specific measures.

Comment 3:

The author should also discuss on the effect of Lung Volume Reduction Surgery on cognitive impairment in suggested interventions (Section 8).

Response 3:

Thank you. I have added the following paragraph in section 8 (page 20; lines 374-381) “Lung volume reduction surgery has been proposed as an intervention for neuropsychological measures. In an older trial, patients with emphysema who had undergone LVRS showed significant improvement in neuropsychological measures (e.g. anxiety and depression) and cognitive functions (e.g. sequential skills and verbal memory) [13]. However, it is to mention the latter study was relitevly small, involving 20 subjects in the control group and 19 patients in the experimental group, meaning that despite the improvement in specific cognitive functions, this area requires in dept investigation. 

Comment 4:

There are numerous typographical errors like “Self-evaluation,” “Emotion” etc in Table 1

Response 4:

Thank you for pointing that out. Spelling was checked and corrected throughout.

Comment 5:

Redundant sentences, for example-lines 29 and 33, lines 220 and 289. Repetition can be avoided. Similarly, Section 3 (line 72) and Section 4 (line 85) can be merged as in both sections association of COPD and cognitive impairment has been described.

Response 5

This has been addressed. However, the 2nd part of the comment (lines 72 and 85) describes the relationship between lung impairment (this could be any lung disease – line 72) and cognitive decline; and the association between COPD (disease specific) and lung impairment – line 85. Thus, I feel this should stay as is. Hope this is acceptable.  

Comment 6

In Reference section, the numbering is appearing twice in each reference.

Response 6

References have been updated and corrected throughout.

Round 2

Reviewer 1 Report

The author still refers to old references but a farther review round is useless.

The other points have been addressed.

Author Response

Comment 1

The author still refers to old references but a further review round is useless.

Response 1:

References are updated as appropriate. However, original studies, even if they are old, were kept. Hope this is acceptable.

Comment 2:

The other points have been addressed.

Response 2:

I am glad you are happy with my responses.

Reviewer 2 Report

The author has tried to address the review comments given earlier. However still some comments are unaddressed or partially addressed which need to be incorporated in the manuscript. 

Please find below the comments.- 

  1. There are typo errors in table 1 which need to be corrected- e.g. “Visuconstruction”, “Self-evalaution” and “emtions”. 
  2. The author has mentioned MMSE, MoCA, ACE, CDT and Mini-Cog test as the tools for assessment of cognitive impairment.  There are several other tools for assessment of cognitive impairment which should be mentioned in the MS e.g. VR-DOT (virtual reality day-out-task), IQCODE (informant questionnaire on cognitive decline in the elderly individuals), CANTAB-PAL (Cambridge Neuropsychological Test Automated Battery-Paired Associated Learning), HDS-R( Hasegawa’s dementia scale-revised), MES (memory and executive screening), etc.

Author Response

Response 1:

This has been addressed.

Comment 2:

The author has mentioned MMSE, MoCA, ACE, CDT and Mini-Cog test as the tools for assessment of cognitive impairment.  There are several other tools for assessment of cognitive impairment which should be mentioned in the MS e.g. VR-DOT (virtual reality day-out-task), IQCODE (informant questionnaire on cognitive decline in the elderly individuals), CANTAB-PAL (Cambridge Neuropsychological Test Automated Battery-Paired Associated Learning), HDS-R( Hasegawa’s dementia scale-revised), MES (memory and executive screening), etc

Response 2:

Thank you. Additional assessment tools were added. Please see Table 3.
